# Protective Immune Signatures Associated with Latent TB Infection in PLHIV—Insights from an Integrative Prospective Immune Monitoring Study

**DOI:** 10.3390/cells14201622

**Published:** 2025-10-17

**Authors:** Shilpa Bhowmick, Pratik Devadiga, Sapna Yadav, Nandan Mohite, Pranay Gurav, Tejaswini Pandey, Varsha Padwal, Namrata Neman, Aarya Suryawanshi, Satyajit Musale, Amit Kumar Singh, Sharad Bhagat, Snehal Kaginkar, Harsha Palav, Shantanu Birje, Shilpa Kerkar, Susan Idicula-Thomas, Vidya Nagar, Priya Patil, Sachee Agrawal, Sushma Gaikwad, Jayanthi Shastri, Nupur Mukherjee, Kiran Munne, Vikrant M. Bhor, Taruna Madan, Vainav Patel

**Affiliations:** 1Viral Immunopathogenesis Lab, ICMR-National Institute for Research in Reproductive and Child Health, Mumbai 400012, India; shilpa_biochem@nirrch.res.in (S.B.); yadavs@nirrch.res.in (S.Y.); mohiten@nirrch.res.in (N.M.); pranayjgurav14@gmail.com (P.G.); wini3959@gmail.com (T.P.); 23varshapadwal@gmail.com (V.P.); nemannamrata511@gmail.com (N.N.); saarya2502@gmail.com (A.S.); musales.nirrch@icmr.gov.in (S.M.); singh.amit489@gmail.com (A.K.S.); bhagats@nirrch.res.in (S.B.); snehal_biochem@nirrch.res.in (S.K.); harshapalav1@gmail.com (H.P.); 2Department of Molecular Immunology and Microbiology, ICMR-National Institute for Research in Reproductive and Child Health, Mumbai 400012, India; iampratikdevadiga@gmail.com (P.D.); bhorv@nirrch.res.in (V.M.B.); 3Department of Child Health research, ICMR-National Institute for Research in Reproductive and Child Health, Mumbai 400012, India; shantanubirje01@gmail.com (S.B.); shilpak124@rediffmail.com (S.K.); munnek@nirrch.res.in (K.M.); 4Biomedical Informatics Centre, ICMR-National Institute for Research in Reproductive and Child Health, Mumbai 400012, India; thomass@nirrch.res.in; 5Department of Medicine, Grant Medical College and JJ Group of Hospitals, Mumbai 400008, India; drvidyanagar@gmail.com (V.N.); patilpriya0312@gmail.com (P.P.); 6Department of Microbiology, TN Medical College and BYL Nair Hospital, Mumbai 400008, India; drsacheeagrawal@gmail.com (S.A.); jsshastri@gmail.com (J.S.); 7Department of Medicine, TN Medical College and BYL Nair Hospital, Mumbai 400008, India; sushmg.sg@gmail.com; 8Department of Cell Physiology and Pathology, ICMR-National Institute for Research in Reproductive and Child Health, Mumbai 400008, India; mukherjeen@nirrch.res.in; 9Department of Innate Immunity, ICMR-National Institute for Research in Reproductive and Child Health, Mumbai 400012, India; guptat@nirrch.res.in

**Keywords:** HIV-TB coinfection, LTBI, HIV pathogenesis, flow cytometry, T-cell activation, PD-1 expression, regulatory T cells, cytokines, intracellular cytokine staining assay, DosR/Rpf

## Abstract

Understanding how HIV-1 pathogenesis affects systemic and TB specific immunity in the setting of latent (LTBI+) compared to active TB infection could provide actionable insights for the prevention of reactivation. Fifty HIV-seronegative and 112 HIV-1-positive anti-retroviral therapy (ART)-naïve participants were stratified as LTBI+ (*n* = 35), active TB+ (*n* = 22) and non-coinfected (*n* = 55) based on an interferon gamma release assay (IGRA) and clinical confirmation prior to receiving TB therapy. Systemic and TB-specific (DosR and Rpf) immune monitoring of cellular subsets, together with multi-analyte plasma analysis, was carried out. Pursuant to isoniazid prophylaxis therapy (IPT) and ART initiation, HIV-1-positive LTBI+ participants (HLTBI+) were followed for up to two years. Before ART initiation, HLTBI+ individuals exhibited the lowest levels of circulating intermediate monocytes, T-cell activation and PD-1 expression, with a decreased frequency of T-regulatory cells and higher circulating IL-10 and IL-17A. PD-1 expression on CD4+ T cell memory subsets, together with opposing anamnestic TNF-α responses to DosR and Rpf, was a discriminatory signature for the HLTBI+ group, as was preserved (following ART) TB-specific TNF-α production, which positively correlated with the CD4/CD8 ratio. Our results highlight an immunomodulatory phenotype conferred by latent TB infection in PLHIV, whose preservation may provide strategies to mitigate TB reactivation.

## 1. Introduction

Most of the 1.5 million new HIV-1 infections in 2024 occurred in low- to middle-income countries [1], of which India currently bears a burden of at least 2.5 million people living with HIV (PLHIV) [2]. Approximately 20–60% of India’s population has a TB infection [3,4,5], and it comprises 26% of all global active TB infections, as well as 26% of the global TB-related deaths [6,7]. Additionally, the risk of reactivation greatly increases (to 5–15% annually) following HIV-1 co-infection [8]. Thus, understanding the pathogenic correlates accompanying both latent and active TB infection, especially in PLHIV, is critical in addressing the public health challenge of co-infection- and reactivation-associated morbidity and mortality.

A previous report on MTB-specific CD4+ T cells showed increased expression of PD-1 in PLHIV with a latent (LTBI) infection and active pulmonary tuberculosis (PTB) compared to HIV-seronegative LTBI and PTB individuals, respectively [9]. Moreover, PLHIV with active TB co-infection exhibit a lower TH17/Treg ratio that decreases with disease progression [10]. Interestingly, in vivo studies have demonstrated that TB reactivation is not necessarily consequent to CD4 depletion or abrogated by anti-retroviral therapy (ART) initiation but, in fact, may depend on the modulation of chronic immune activation, along with the altered effector T-cell phenotypes and dysregulated T-cell homeostasis [11,12]. Additionally, responses in PLHIV, with and without TB co-infection, against Rpf—a TB antigen that has been shown to be necessary for reactivation [13,14]—have not been studied.

Our study undertook the delineation of unique systemic and TB-specific pathogenic signatures associated with LTBI and active TB in the context of ART-naïve and -receiving HIV-1 patients towards developing actionable targets for strategies to minimize morbidity and mortality caused by TB reactivation. We demonstrate a unique immunomodulatory phenotype, irrespective of ART, in PLHIV with latent infections that could possibly protect against reactivation if preserved.

## 2. Materials and Methods

### 2.1. Study Approval

The study protocol was approved by the ICMR-NIRRCH Ethics Committee (Ethics No. 348/2018, 410/2020), ART Center of JJ Group of Hospitals (No.: IEC/Pharm/RP/183/Oct/2020), ECARP BYL Nair Hospital (ECARP/2020/152 and ECARP/2020/109) and NACO. Study participants were recruited from JJ Hospital and B.Y.L. Nair Charitable Hospital, Mumbai, in accordance with the approved protocol. Signed informed consent documents were obtained from the study participants prior to sample collection.

### 2.2. Study Participants

Based on the only prior study from India [15] (Rakshit et al., 2020) that had investigated the immune monitoring of TB-specific responses in individuals with HIV-LTB co-infection using a relatively small cohort (9–10 individuals per group), we designed our study to be more robust and extend these findings by recruiting a larger and more stratified cohort. HIV-1-positive and HIV-seronegative individuals were serially screened and categorized into LTBI+ and active TB+ groups based on IGRA results, along with clinical confirmation of active TB using chest imaging, sputum smear microscopy and/or GeneXpert testing. 

The HLTBI+ group served as the primary test group and was followed longitudinally for 2 years at 6-month intervals to monitor for TB reactivation. The remaining groups served as baseline comparators to assess systemic and antigen-specific immune responses across different infection statuses. After the screening and recruitment of at least 30 individuals in the HTLBI+ group, sufficient to perform extensive flow cytometry and other assays (absolute counts, whole-blood immunophenotyping, ICCSA, ELISA and Luminex) and detect statistically significant differences in immune parameters between groups using ANOVA, we stopped the cross-sectional recruitment and followed up with these individuals for a period of 2 years post-initiation of ART. One hundred and twelve HIV-1-positive ART-naïve and 50 HIV-seronegative individuals were recruited, and peripheral blood was collected in EDTA and lithium heparin vacutainers after obtaining signed informed consent from the recruited participants. The recruited participants were screened by HIV Tridot to confirm HIV-1 seropositivity. Blood collected in lithium heparin tubes was used to stratify individuals further into different groups based on their LTBI and active TB status, which was performed via the interferon gamma release assay (IGRA) using the QuantiFERON-TB Gold Plus test to confirm LTBI at NIRRCH. The assay was performed using one replicate per sample, except for standards, which were run in duplicates, as per the manufacturer’s recommendation. Active TB was reported by the hospital through a chest scan, sputum and/or smear positivity and GeneXpert positivity. HIV+ individuals with LTBI+ co-infection were followed for a period of up to 2 years at intervals of 6 months following the initiation of anti-retroviral therapy (ART). Time point 0 (TP0) indicates cross-sectional sampling, TP1 indicates the time point of blood collection after 6-8 months of ART (*n* = 17), TP2 indicates 12–14 months of ART (*n* = 11), TP3 indicates 18-20 months of ART (*n* = 8) and TP4 indicates 24–26 months of ART (*n* = 7). Whole blood was collected at every time point to perform assays.

### 2.3. Assays for Absolute Cell Counts

Peripheral whole blood collected in fresh ethylenediaminetetraacetic acid (EDTA) tubes was used to determine the absolute numbers of CD4+ and CD8+ T cells, B cells, NK cells and monocytes using the stain/lyse/no wash technique. Briefly, 50 µL of whole blood was stained with the following antibodies: anti-CD45-FITC (BD Biosciences, San Jose, CA, USA; clone: 2D1; catalog no.: 347463), anti-CD3-APC-Cy7 (clone: SK7; catalog no.: 557832), anti-CD4-BV480 (BD Biosciences, San Jose, CA, USA; clone: RPA-T4; catalog no.: 566104), anti-CD8-BV605 (BD Biosciences, San Jose, CA, USA; clone: SK1; catalog no.: 564116), anti-CD19-BV421 (BD Biosciences, San Jose, CA, USA; clone: HIB19; catalog no.: 562440), anti-CD16-PE (BD Biosciences, San Jose, CA, USA; clone: 3G8; catalog no.: 555407), anti-CD56-BV786 (BD Biosciences, San Jose, CA, USA; clone: NCAM16.2; catalog no.: 564058) and anti-CD14-Pe-Cy7 (BD Biosciences, San Jose, CA, USA; clone: M5E2; catalog no.: 557742). RBCs were lysed after 20 min of incubation with antibodies, followed by adding 50 µL of BD liquid counting beads, and data acquisition was carried out on the BD FACS Aria Fusion^TM^ (BD Biosciences, Franklin Lakes, NJ, USA) and BD Symphony A3 (BD Biosciences, Franklin Lakes, NJ, USA). A total of 20,000 events in the bead population were acquired. Data analysis was performed using the FlowJo^TM^ v10.10 software. 

### 2.4. Plasma Viral Load Estimation

Viral nucleic acid was isolated manually from plasma samples using the QIAmp Viral RNA Mini Kit (Qiagen, Venlo, The Netherlands; catalog no: 52904). Viral load (copies per milliliter) was quantified using the AltoStar^®^ HIV RT-PCR Kit 1.5 (Altona Diagnostics, Hamburg, Germany; catalog no.: AS0221513), with a detection limit of 34 copies/mL of plasma.

### 2.5. Whole-Blood Immunophenotyping

Whole-blood immunophenotyping was carried out as described earlier [16,17]. Briefly, 200 µL of whole blood was processed using the stain/lyse/wash technique with monoclonal antibodies conjugated with fluorochrome to determine the activation and exhaustion status of T-cell subsets of both CD4+ and CD8+ T cells using anti-CD3-APC-Cy7 (clone: SK7; catalog no.: 557832), anti-CD4-BV480 (clone: RPA-T4; catalog no.: 566104), anti-CD8-BV605 (clone: SK1; catalog no.: 564116), anti-CD28-Pe-Cy7 (BD Biosciences, San Jose, CA, USA; clone: 28.2; catalog no.: 560684), anti-CD45-RA-Per-CP-Cy5.5 (BD Biosciences, San Jose, CA, USA; clone: HI100; catalog no.: 563429), anti-CCR7-PECF594 (BD Biosciences, San Jose, CA, USA; clone: 150503; catalog no.: 562381), anti-HLADR-BV421 (BD Biosciences, San Jose, CA, USA; clone: L243; catalog no.: 568649), anti-CD38-BUV395 (BD Biosciences, San Jose, CA, USA; clone: HIT2; catalog no.: 740294) and anti-PD-1-PE (BD Biosciences, San Jose, CA, USA; clone: EH12.1; catalog no.: 560795). Frequencies of Tregs were determined using anti-CD25-Pe-Cy7 (BD Biosciences, San Jose, CA, USA; clone: M-A251; catalog no.: 557741) and anti-CD127-AF647 (BD Biosciences, San Jose, CA, USA; clone: HIL-7R-M21; catalog no.: 558598). Acquisition was carried out on the BD FACS Aria Fusion^TM^ and a BD Symphony A3 flow cytometer, and data analysis was carried out using the FlowJo^TM^ v10.10 software. Populations were gated using unstained and fluorescence minus one (FMO) controls. A total of 1.5 lakh events in the CD3+ T-cell gate were acquired.

### 2.6. ELISA and Luminex Assay

Peripheral whole blood was centrifuged at 400× *g* and plasma was separated, followed by flash freezing in liquid nitrogen and storage at −80 °C until use. Samples were thawed for the batch analysis of plasma analytes sCD14, sCD163, C-reactive protein (CRP), D-dimer, IL-10, IP-10, IL-17A, IL-12p70 and IFN-γ. sCD14, sCD163 and CRP were measured by ELISA using the Human Soluble Cluster of Differentiation 14 ELISA Kit (Mybiosource, San Diego, CA, USA; catalog no.: MBS763332), Human CD163 Molecule ELISA Kit (Mybiosource, San Diego, CA, USA; catalog no.: MBS702685) and Human C-Reactive Protein CRP Quantikine ELISA Kit (R&D Systems, Minneapolis, MN, USA; catalog no.: DCRP00B), respectively. The remaining analytes were measured using the Luminex assay kit PPX-06 PROCARTAPLEX 6 PLEX-1PLATE 6-PLEX (PPX-06) (Invitrogen, Waltham, MA, USA; catalog no.: PPX-06-MXFVNUT). The samples were acquired using the Luminex xMAP Intelliflex version 1.1 and Bio-Plex^®^ 200 System version 6.2. All assays were run in duplicate to ensure the accuracy and reproducibility of the cytokine measurements.

### 2.7. Intracellular Cytokine Staining Assay (ICCSA)

The ICCS assay was performed using cryopreserved PBMCs, as described earlier in [18,19], for both cross-sectional and follow-up samples. Cryopreserved PBMCs were thawed and resuspended in complete media [RPMI-1640 (Himedia, Thane, India; catalog no: AL028) medium supplemented with 10% FBS (Himedia, India; catalog no: RM9955) and 1% penicillin/streptomycin(Gibco, Waltham, MA, USA; catalog no: 10378016)], followed by 1 h of resting in resting media (complete media supplemented with DNase I [1 mg/mL]) at 37 °C with 5% CO_2_ conditions. Rested PBMCs were then either left unstimulated or stimulated with a TB latency-specific pool of 4 DosR regulon-encoded proteins (Rv1737, Rv1733, Rv2628 and Rv2029) and Rpf proteins (Rv0869 and Rv2389) at a final concentration of 10 μg/mL, produced by Kees LMC Franken as recombinant proteins at the Department of Infectious Diseases, Leiden University Medical Center, and in the presence of a pool of 15 mer peptides overlapping by 11 amino acids of ESAT-6/CFP-10 (EC) at a final concentration of 1 μg/mL, for a total of 16 h, in the presence of anti-CD107a mAb conjugated with BV786 to detect degranulation. This choice of the DosR and Rpf protein pool was based on robust evidence from multiple independent studies demonstrating that these specific antigens induce the strongest IFN-γ responses among all members of their respective regulons in MTB-exposed individuals [20,21,22]. PBMCs were also stimulated in the presence of phorbol 12-myristate 13-acetate (PMA) (50 ng/mL) (Invitrogen, Waltham, MA, USA) and ionomycin (1 μg/mL) (Invitrogen, Waltham, MA, USA) as positive controls, and PBMCs incubated with media alone were the negative controls. Brefeldin (Golgi plug, 1 μL/mL) (BD Biosciences, San Jose, CA, USA; catalog no.: 555029) and monensin (Golgi stop, 0.7 μL/mL) (BD Biosciences, San Jose, CA, USA; catalog no.: 554724) were added after an hour of stimulation for cells stimulated with PMA and EC and after 4 h for cells stimulated with latency proteins [23]. After stimulation, PBMCs were washed with PBS (Gibco™, Waltham, MA, USA; catalog no. 10010023), followed by live/dead staining for 30 min in the dark with the LIVE/DEAD^TM^ Fixable Violet Dead Cell Stain Kit (Invitrogen, Waltham, MA, USA; catalog no.: L34963). Cells were then washed with stain buffer (PBS containing 0.2% FBS), followed by staining with surface antibodies. Cells were then fixed and permeabilized using BD Cytofix/Cytoperm^TM^ (BD Biosciences, San Jose, CA, USA; catalog no.: 554714) for 30 min at RT, followed by the intracellular staining of cytokines using specific mAbs. Next, cells were washed and resuspended in 300 μL of stain buffer. Stained samples were acquired on the BD FACS Aria Fusion^TM^ and BD Symphony A3 flow cytometer. A total of 2–2.5 lakh events in the live T-cell gate were acquired. Responses ≥ 0.02% with more than 20 events above the background value were positive. 

### 2.8. Antibodies for ICCS Assay

The following panel of antibodies was used to define the anamnestic T-cell response against MHC class I and class II restricted epitopes, due to previous exposure: anti-CD3-APC-Cy7 (clone: SK7; catalog no.: 557832), anti-CD4-BV480 (clone: RPA-T4; catalog no.: 566104), anti-CD8-BV605 (clone: SK1; catalog no.: 564116), anti-IFN-γ-AF488 (BD Biosciences, San Jose, CA, USA; clone: B27, catalog no.: 557718), anti-IL-10-AF647 (BD Biosciences, San Jose, CA, USA; clone: JES3-9D7; catalog no.: 501412), anti-MIP-1β-PE (BD Biosciences, San Jose, CA, USA; clone: D21-1351; catalog no.: 550078), anti-TNF-α-PECy7 (BD Biosciences, San Jose, CA, USA; clone: MAb11; catalog no.: 557647) and anti-CD107a-BV786 (BD Biosciences, San Jose, CA, USA; clone: H4A3; catalog no.: 563869).

### 2.9. Statistics

The FlowJo^TM^ software (version 10.10) was used to analyze data generated using flow cytometry, and the GraphPad Prism software version 10 was used to perform statistical analysis. The data were represented as scatter plots, violin plots and bar graphs indicated with median values. Comparison between groups was performed by the Kruskal–Wallis one-way ANOVA non-parametric test. Pairwise comparisons between matched samples were carried out using either the Friedman one-way ANOVA non-parametric test or Wilcoxon matched-pairs signed-rank test. Bivariate associations were determined by Spearman’s rank correlation test. For all statistical calculations, *p* < 0.05 was considered significant. R studio was used to perform the hierarchical clustering analysis.

## 3. Results

### 3.1. Demographic, Immunological and Virological Characteristics of Study Population

Out of a total of 162 individuals recruited for this study, 112 were therapy-naïve HIV-1-positive and 50 were HIV-seronegative controls (Table 1 1 and Appendix A). PLHIV were further classified as LTBI−positive (HLTBI+ [n = 35]) or -negative (HLTBI− [*n* = 55]) and active TB-positive (HTB+ [*n* = 22]). Similarly, seronegative controls were stratified into LTBI+ (n = 17), LTBI− (*n* = 30) and TB+ (*n* = 3) groups. As described in Table 2 no significant difference was observed in age across PLHIV groups. The majority of the individuals in the HIV-1-positive groups were male (*n* = 77; 68.75%). All HIV-positive groups had significantly lower absolute CD4 counts and CD4/CD8 ratios compared to the HIV-seronegative group (Appendix A). Interestingly, within the HIV-positive groups, we observed the highest absolute CD4 count (*p* = 0.0014 compared to HTB+) and CD4/CD8 ratio (*p* = 0.0030 compared to HTB+) in the HLTBI+ group. Conversely, viremia was highest (significantly, *p* = 0.0273 compared to HLTBI− and *p* = 0.0357 compared to HLTBI+) in the HTB+ group, suggesting that co-infection with active TB favored HIV-1 viral replication and concomitantly increased pathology in terms of lowered CD4 counts and CD4/CD8 ratios (Appendix A). Overall, viral loads negatively correlated with absolute CD4 counts and CD4:CD8 ratios (Appendix A). Among other cellular subsets (Appendix A), the B-cell count was observed to be significantly lower in PLHIV groups compared to HIV-seronegative groups, with the exception of the TB+ group (Figure 1A). NK cell counts were significantly higher in HLTBI+ individuals compared to the HLTBI− (*p* = 0.0004) and HTB+ (*p* = 0.0089) groups (Figure 1B), and further resolution into subsets (Figure 1D–F) resulted in the identification of these as mainly CD16+ NK cells. Monocyte counts (Figure 1C), overall, trended towards being lower in PLHIV; however, interestingly, HLTBI+ individuals showed a significant reduction (similar to HIV-negative groups) in intermediate (CD14++CD16+) monocytes, compared to the non-co-infection (HLTBI−) group (*p* = 0.0388); these are known to be otherwise elevated in HIV infection (Figure 1H).

Based on the expression of pan-leukocyte markers, i.e., CD45, we then further gated these populations and performed an unsupervised dimensionality reduction analysis by t-Stochastic Neighbor Embedding (tSNE), followed by a graph-based clustering analysis (XShift) (Figure 1J–M). A total of 10 clusters were derived, and two each were identified to be differentially occurring in the TB co-infect groups. Clusters 2 and 10, representing subsets with high CD14+ expression, were contributed to largely by the HTB+ group, whereas Clusters 8 and 9 were largely composed of cells from the HLTBI+ group, wherein the former included CD4+CD8+ T cells with apparent CD14 expression, possibly representing monocyte–T cell complexes. Cluster 9 consisted of non-T cells expressing high levels of CD8, CD14 and CD16. Interestingly, Cluster 4 was underrepresented in both TB co-infected groups and consisted of cells with high expression of CD3, CD4 and CD8.

### 3.2. Lower T-Cell Activation and PD-1 Expression in HLTBI+ Individuals

CD4+ and CD8+ T-cell activation was evaluated by measuring the co-expression of CD38 and HLA-DR in naïve, memory and effector subsets of T cells (Appendix A). As expected, and as observed in Figure 2, activation levels were observed to be higher in HIV-positive groups compared to HIV-seronegative groups in all subsets of both CD4+ and CD8+ T cells, except for the CD4+ T-cell naïve subset. Interestingly, significantly lower levels of activation in the HLTBI+ (CM *p* = 0.0171, TM *p* = 0.0313) group compared to the HTB+ group was observed in CD4+ subsets, whereas these levels were significantly lower in CM (*p* = 0.0118) and TM (*p* = 0.0428) CD8+ T cells compared to the HLTBI− group (Figure 2B,C,F,G), suggestive of the systemically lower activation of memory T cells in PLHIV with LTBI. However, this signature was not observed for EM subsets, wherein PLHIV with an active TB infection had lower activation levels compared to the other two HIV-positive groups (Figure 2D,H). Next, we examined the frequency of circulating PD-1 expressing potentially dysfunctional T cells in our study groups (Appendix A). As depicted in Figure 3B,C, the expression of PD-1 was observed to be significantly higher in the HLTBI− and HTB+ groups in the CM and TM subsets of CD4+ T cells, respectively, compared to that in the HIV-seronegative groups (LTBI− and LTBI+). Intriguingly, we observed that the frequency of memory CD4+ PD-1+ T cells was lower (significantly for CM and TM subsets) in the HLTBI+ group compared to either the non-co-infect group (CM *p* < 0.0001, TM *p* < 0.0001) or that with active TB (CM *p* < 0.0001, TM *p* < 0.0001), and these levels were found to be comparable to those in the HIV-negative groups. No significant difference was observed in the levels of PD-1 between groups for the naïve subset of CD4+ T cells (Figure 3A). In the case of CD8+ T cells, an overall increase in PD-1 expression in all subsets was observed in PLHIV with and without co-infection compared to HIV-seronegative groups. Interestingly, the CM subset in HLTBI+ individuals showed lower levels of PD-1 expression than the other two HIV+ groups, especially when compared (significantly) to HLTBI− individuals (*p* = 0.0339). However, this trend was not observed in the case of other subsets (Figure 3E–H).

On performing tSNE and XShift gating analysis on CD3+ T cells, we observed the presence of unique cell types specific to our groups (Figure 3I–L), where, out of 10 clusters, Clusters 10, 5 and 2, which were primarily CD8+ T-cell subsets expressing low to moderate HLADR, accounted for the majority of events and were equally contributed by all three HIV+ groups. CD8+ naïve T cells expressing high levels of HLADR (Cluster 1) were observed to be completely absent in HTB+ individuals and expressed mostly by HIV-seronegative individuals, followed by HLTBI+ individuals. The HLTBI+ group was the highest contributor for central memory T cells (Cluster 3), along with central memory CD4+ T cells expressing PD-1 (Cluster 4), highlighting the preservation of these cell types in the HLTBI+ group. As opposed to Clusters 3 and 4, Clusters 7 and 8 (activated CD4+ T-cell subsets) were underrepresented in the HLTBI+ group, supporting our observation of low activation within this group. The low levels of these clusters in the HTB+ group may have reflected the low CD4 counts overall in these individuals.

### 3.3. Reduced Frequency of Regulatory T Cells in HLTBI+ Individuals

To assess Treg dysfunctionality in our cohort, we evaluated the circulating levels of Tregs (CD3+ CD4+ CD25^high^ CD127^low^) (Figure 4 and Appendix A). As depicted in Figure 4A, the frequency of CD4+ Tregs was elevated following HIV-1 infection compared to HIV-seronegative individuals, except, notably, in the case of the HLTBI+ group. In fact, these levels were clearly reduced in HLTBI+ individuals compared to HLTBI− individuals (*p* = 0.0141), while the Treg frequency showed a stark and significant increase in HTB+ individuals (*p*< 0.0001), who also had the highest viral loads and poorest disease progression markers (Table 1 and Appendix A). Deeper functional analysis revealed an overall increase in the frequency of memory Tregs within HIV-positive individuals, reflective of increased activation and dysfunction, with a concomitant decrease in the naïve Treg frequency (Figure 4B,C). The unique signature associated with the HLTBI+ group was extended to this analysis, where this group of individuals exhibited a higher frequency of naïve and lower frequency of memory Tregs among HIV-positive individuals. The enumeration of the absolute Treg counts demonstrated that all HIV-positive groups had lower counts of total, naïve and memory Tregs, reflective of overall CD4 depletion (Figure 4D–F and Appendix A). Here, too, the HLTBI+ group showed the lowest drop in counts compared to HIV-seronegative individuals. Taken together, and recapitulating disease progression (Appendix A), the Treg count correlated positively with the CD4 count, while the Treg frequency was negatively correlated with the CD4 count, contributed mainly by the memory Treg subset.

An unsupervised analysis of Treg populations was performed, which generated a total of five clusters, all of which had similar numbers of contributing cells. Cluster 1 (memory Treg-like) was underrepresented in both the HIV-seronegative and HLTBI+ groups. Conversely, Clusters 3 and 4, resembling naïve T cells, were underrepresented in the HTB+ group.

### 3.4. Plasma Analyte Analysis

In addition to assessing the systemic levels of surface markers, we also evaluated the circulating concentrations of surrogate indicators of inflammation, microbial translocation, monocyte/macrophage activation, gut integrity and TB disease status. Although these differences were not statistically significant, we observed notable trends among the HIV-positive groups (Figure 5 and Appendix A). Specifically, the sCD14 and sCD163 levels tended to be elevated in HIV-positive individuals compared to HIV-seronegative controls (Appendix A). D-dimer, a potential biomarker for TB disease, was detected at higher levels in the HTB+ group (Appendix A). Interestingly, the IL-10 (*p* = 0.0814) and IL-17A (*p* = 0.0471) levels were increased in the HLTBI+ group, especially compared to the HTB+ group (Figure 5A,B). In contrast, the IP-10 levels were low in both the HLTBI− (*p* = 0.0061) and HLTBI+ (*p* = 0.0522) groups compared to the HTB+ group (Appendix A). No distinct pattern was observed for IL-12p70 (Appendix A), and the IFN-γ levels were largely undetectable in most individuals (Appendix A). 

### 3.5. Identification of HLTBI+-Specific Systemic Immune Signatures

As depicted in Figure 5C–E, correlation matrices for each study group of assessed systemic immune markers were generated to evaluate group-specific signatures. Classical HIV disease progression markers, i.e., the CD4 count and CD4:CD8 ratio, showed strong negative correlations with viral loads and functional signatures, such as PD-1 expression in the CM and TM compartments of CD4+ T cells, in the HLTBI− and HTB+ groups, which were not significant for the HLTBI+ group. Moreover, the HTB+ group uniquely showed a significant negative correlation between these markers and circulating IL-17A. A significant positive association between the Treg frequency and PD-1 expression in CM and TM CD4+ T cells was observed in the HLTBI− (CM *r*= 0.3086, *p* = 0.0260; TM *r* = 0.1473, *p* = 0.2973) and HTB+ (CM *r* = 0.2492, *p* = 0.2635; TM *r* = 0.5164, *p* = 0.0139) groups, whereas an opposing significant negative association was observed in the HLTBI+ (CM *r* = −0.4032, *p* = 0.0181; TM *r* = −0.4747, *p* = 0.0046) group, suggestive of a distinct functional relationship between the modulation of T-cell activation and Tregs in this group. HLTBI+ individuals alone (among the HIV+ groups) displayed a unique negative association between PD-1 expression in CD4+ subsets (TM and CM) and the circulating levels of IL-10 (CM *r* = −0.4487, *p* = 0.0362; TM *r* = −0.4161, *p* =0.0541), IL-12p70 (CM *r* = −0.4500, *p* = 0.0822; TM *r* = −0.4353, *p* = 0.0937) and IL-17A (CM *r* = −0.4374, *p* = 0.1198; TM *r* = −0.4110, *p* = 0.1458). This association extended the above-described unique signature of low PD-1 and higher IL-10 and IL-17A levels (compared to other HIV+ groups) observed in HLTBI+ individuals. Additional unique signatures associated with the HLTBI+ group were the strong negative association of CRP levels with monocyte counts and the significant positive associations of B-cell counts with monocyte and NK counts. Finally, the strong positive correlations between the NK and monocyte subset counts observed in the non-co-infect and HLTBI+ groups were absent in the HTB+ group. 

### 3.6. PD-1 Expression on CD4+ Memory T Cells: An Independent Discriminatory Marker 

To determine whether any of the signatures described above were capable of independently segregating the three HIV+ groups, hierarchical clustering was employed (Figure 5F and Appendix A). PD-1 expression on CD4+ CM and TM subsets apparently segregated nodes with high heat, comprising mainly HLTBI− and HTB+ individuals, and the ones with the lowest heat included only HLTBI+ individuals. The intermediate-heat region comprised HLTBI− and HLTBI+ individuals, reflecting our hypothesis of the HLTBI+ and HTB+ groups representing two extremes of immune pathogenesis (Figure 5D). Furthermore, an outlier analysis was carried out on four individuals from the HLTBI+ group (marked with arrows), which seemed to distribute within the area of highest heat. In terms of the available data pertaining to disease progression markers (Appendix A), it was observed that all four outliers lay in the 50–75th or 75–100th percentiles of their groups with respect to the activation of CD4+ and CD8+ T-cell memory subsets. Moreover, in terms of PD-1 expression on CD8+ T-cell memory subsets, as well as the viral load, at least two of the four individuals were distributed in the >50th percentile.

### 3.7. Response to Anti-Retroviral Therapy (ART) in HLTBI+ Individuals: Impaired Immune Restoration

Upon the initiation of ART, followed for up to 2 years, as shown in Appendix A, and as expected, standard HIV disease progression markers such as absolute CD4 counts, the CD4/CD8 ratio and T-cell activation all showed significant, albeit impaired, restoration that was subset-dependent in the context of virological suppression. These observations were supported by matched data that were available for a smaller set of participants (Appendix A–J). The expression of PD-1 by CD4+ T-cell subsets was not significantly altered, reflective of the near-normal levels of expression observed in this group prior to ART (Figure 6A–D). This contrasted with the CD8+ T-cell compartment (Figure 6E–H and Appendix A), where the restoration of PD-1 expression was dramatic, as also observed in a smaller, matched, longitudinally followed cohort (Appendix A). Contrary to the restorative trends observed in some T-cell subsets of activation and PD-1 expression, a persistently elevated frequency of memory Tregs was observed throughout the follow-up period, indicating viral reservoir-driven activation that was not completely abrogated by ART (Figure 6I–K and Appendix A). Longitudinal plasma analysis in the HLTBI+ group, where available, also provided evidence for some immune-restorative effects of ART, demonstrated through clear drops in the circulating D-dimer (*p* = 0.0156) and sCD163 (*p* = 0.0273) levels by 6 months, as well as increases in circulating IL-12p70 (*p* = 0.0156) levels. Interestingly, the IL-10 and IL-17A levels, which trended higher in this group prior to ART, did not show a significant decrease at 6 months post-ART (Appendix A).

### 3.8. TB-Specific Responses in ART-Naïve PLHIV with Co-Infection

We evaluated TB-specific cellular immunity in our cohort using ex vivo stimulation with the latency-associated proteins dormancy survival regulon (DosR) and, for the first time, resuscitation-promoting factor (Rpf) (Appendix A). As shown in Figure 7A,B and Appendix A, the DosR responder rates (RRs) were highest in the HTB+ group for IL-10, Mip-1β and CD107a. Interestingly, responses were also detected in the LTBI−negative groups, and the HIV infection status seemed to affect the RRs for Mip-1β, CD107a and TNF-α, which, except for the latter, were all higher in positive individuals. With respect to Rpf responses, our results demonstrated the highest RRs in the HTB+ group across all five intracellular markers estimated, especially for IL-10 and Mip-1β. Notably, within the HTB+ and HLTBI+ groups, opposing RRs for DosR and Rpf were observed in terms of TNF-α. The responder rates for Rpf were the highest, and those against DosR were the lowest, in the HTB+ group, with the opposite pattern observed for the HLTBI+ group. Next, we scored the number of antigen-specific functional markers detected against these antigens for our cohort (Figure 7C,D) and observed the preponderance of three function responses, across CD4+ and CD8+ T cells, in the co-infect groups for DosR. While the HIV infection status did not seem to affect this number with respect to Rpf responses, a notable exception was the HTB+ group, where four function responses were noted in both T-cell compartments. A comparative analysis of antigen-specific functional markers (Figure 7E,F), in the case of DosR, revealed that the dominant response was an IL-10, Mip-1β or IL-10, Mip-1β, TNF-α response within the HIV-negative and -positive groups, respectively. Moreover, a unique two-function (IL-10, TNF-α) and three-function (IL-10, Mip-1β, CD107a) response was observed against DosR in the HIV- and HIV+ groups, respectively. Furthermore, the CD8+ T-cell response in HIV+ individuals consisted significantly of three functions, Mip-1β, CD107a and TNF-α cells, not observed in CD4+ T cells or in HIV-negative individuals. The delineation of Rpf-specific responses highlighted a four-function (IL-10, Mip-1β, CD107a, TNF-α) CD8+ T-cell antigen-specific response and a four-function, non-cytotoxic (IFN-γ, IL-10, Mip-1β, TNF-α) CD4+ T-cell response, uniquely in the HTB+ group. Additionally, the apparent contraction of Rpf-specific CD8+ T-cell responses in HIV+ individuals was observed, from three functions (IL-10, Mip-1β, CD107a and IL-10, Mip-1β, TNF-α) to two functions (Mip-1β, CD107a and IL-10, TNF-α, respectively).

### 3.9. TB-Specific Responses in PLHIV with LTBI Following Initiation of ART

The impact of ART initiation on the above-described TB-specific responses was assessed at up to 14 months in the HLTBI+ group. As shown in Figure 8A–D, and also apparent when the quantum of responses was examined (Appendix A), we observed transient increases (at TP1) in RRs for both antigens across the CD4+ and CD8+ T-cell compartments, except in the case of Mip-1β, which showed a consistent decline. Similarly, the IFN-γ RRs declined for both antigens in CD8+ T cells, and the TNF-α RRs declined only against Rpf in this compartment. While the RRs of most functions declined after TP1, a notable exception was the RR observed against Rpf for the cytolytic marker CD107a in CD4+ T cells following extended ART. When evaluating (Figure 8E,F) the average number of markers detected and the composition of these responses against both antigens for the time point where the most data were available (TP1), we observed no difference in the former and the profile reported above (Figure 7C,D) for the ART-naïve time point (TP0). However, when the composition of responses was assessed, in the case of CD4+ T cells, for both antigens, a clear contraction in the three-function (IL-10, Mip-1β, CD107a) response was observed, with the corresponding expansion of another three-function (IL-10, CD107a, TNF-α) response. While the post-ART response against DosR within CD4+ T cells seemed to include a de novo three-function profile (Mip-1β, CD107a, TNF-α), this signature contracted in the context of Rpf responses following ART. For both antigens, another common signature observed was an increase in three-function responses (IFN-γ, IL-10, TNF-α) following ART. With respect to the CD8+ T-cell compartment, the unique two-function profiles observed against Rpf in the therapy-naïve setting remained largely unaltered at 6 months (TP1) following therapy initiation. However, clear alterations in these profiles were observed for DosR, which included contractions in the IFN-γ, Mip-1β, TNF-α profile and the expansion of the IFN-γ, IL-10, CD107a profile.

### 3.10. Integrated Pathogenic Signatures Delineating Latent and Active TB Co-Infection

In this study, we had the opportunity to assimilate data from both systemic and TB-specific immune responses in PLHIV that were clinically disparate in terms of TB co-infection and HIV disease progression markers. Indeed, as shown in Figure 9, correlation matrices of both types of immune data yielded signatures that defined HIV-1 pathogenesis in HLTBI+ and HTB+ individuals distinctly. Two such determinants were the CD4 count (*r* = 0.4847, *p* = 0.0223) and CD4/CD8 ratio (*r* = 0.4832, *p* = 0.0309), which were positively correlated with DosR-specific TNF-α production by CD8+ T cells, observed only in the HLTBI+ group. Conversely, the strong negative correlations between the CD4 count and cytolytic responses against DosR by CD4+ T cells that were observed in the other two HIV-positive groups were absent in the HLTBI+ group. Interestingly, however, this response was strongly and negatively correlated with the viral load (*r* = −0.4548, *p* = 0.0383) only in the HLTBI+ group, suggesting a protective role in blunting HIV replication. Additionally, DosR-specific IL-10 production by CD8+ T cells was strongly and negatively correlated with activation on naïve CD4+ T cells (*r* = −0.6730, *p* = 0.0006), indicative of a suppressive milieu accompanying HLTBI co-infection. Moreover, a strong negative correlation between DosR-specific TNF-α (*r* = −0.4297, *p* = 0.0460) and IFN-γ (*r* = −0.7492. *p* < 0.0001) production by CD4+ T cells and PD-1 expression on naïve CD8+ T cells, uniquely observed in the HLTBI+ group, highlighted the probable protective Th1 responses within these individuals. Similarly, a strong positive correlation between DosR-specific IFN-γ production by CD8+ T cells and the activation of the CD8+ EM subset (*r* = 0.5387, *p* = 0.0118) was observed only in this group. With respect to the active co-infection HTB+ group, a DosR-specific cytolytic response by CD4+ T cells was strongly and positively correlated with the acquisition of PD-1 on CD4+ TM cells (*r* = 0.8467, *p* = 0.0036), as well as the Treg frequency (*r* = 0.7852, *p* = 0.0115), probably consequent to heightened immune activation and pathogenesis. Furthermore, a putative protective response, probably arising in the later stages of active TB, was observed in terms of a negative correlation of the DosR-specific TNF-α response by CD8+ T cells with PD-1 expression on CD8 EM cells (*r* = −0.7668, *p* = 0.0238).

Unique correlations with Rpf responses, usually associated with an active (reactivated) TB state, were, as expected, mainly observed in the HTB+ group. Most of these were apparently protective TNF-α responses produced by either CD4+ or CD8+ T cells, which were strongly and negatively correlated with CD4 CM (CD4+ TNF-α *r* = −0.7374, *p* = 0.0126; CD8+ TNF-α *r* = −0.5688, *p* = 0.0735) and CD8 NV (CD4+ TNF-α *r* = −0.07005, *p* = 0.0201; CD8+ TNF-α *r* = −0.7156, *p* = 0.0173) activation. A single strong, positive association between the Rpf-specific IFN-γ response by CD4+ T cells and CD4+ NV activation (*r* = 0.9289, *p* = 0.0009) was observed, suggesting its linkage to disease progression. Interestingly, we noted heretofore unreported negative associations between Rpf-specific cytolytic CD4+ T-cell responses and the PD-1 expression of TM (*r* = −0.3624, *p* = 0.0750) and EM (*r* = −0.5251, *p* = 0.0070) CD8+ T cells in the HLTBI+ group, which may have also been protective.

### 3.11. Preservation of Protective TB-Specific Signatures Following ART Initiation

In this cohort, we were able to track the dynamics of the aforementioned integrated signatures for up to 6 months post-ART initiation in 50% (*n* = 17) of the HLTBI+ group. At the outset, with respect to DosR (Figure 10A), we noted that TNF-α production by both CD4+ and CD8+ T cells and its associations with key markers of HIV-1 disease progression, such as CD4 counts, the CD4/CD8 ratio and levels of activation and PD-1 expression, on T cells was maintained. Indicative of the partially immune-restorative effect of ART and CD4+ T-cell rebound, we observed de novo signatures such as the acquisition of cytolytic responses and the production of IFN-γ in CD8+ T cells that were negatively correlated with activation and PD-1 expression within the CD8+ T-cell compartment. Interestingly, DosR-specific IL-10 production by CD4+ T cells, negatively correlating with activation and PD-1 expression in CD8+ T cells, was also observed uniquely following ART in these individuals. Conversely, some signatures, such as cytolytic CD4+ T-cell responses negatively correlating with the viral load, were lost following therapy. Additionally, IFN-γ production in CD4+ T cells and IL-10 production by CD8+ T cells, both negatively correlating with PD-1 expression and activation, respectively, within the naïve CD4+ T-cell subset were absent at 6 months post-ART initiation.

In the case of Rpf specific responses (Figure 10B), where such an analysis was carried out for the first time, it was noted that, as for DosR, the negative correlation of TNF-α production in both CD4+ and CD8+ T cells with PD-1 expression on memory T cells was preserved, along with a positive association of the CD4/CD8 ratio with TNF-α production by CD8+ T cells (*r* = 0.3481, *p* = 0.1857). However, the majority of responses that strongly correlated with T-cell functional markers were de novo responses, including the acquisition of cytolytic activity in the CD8+ T-cell compartment, which negatively correlated with PD-1 expression on naïve and memory T-cell subsets. Interestingly, the increase in the Rpf-fspecific CD8+ T-cell cytolytic potential seemed to occur concurrently with the loss of this feature, together with IFN- γ production, in the CD4+ T-cell compartment prior to therapy initiation. This suggests a role for the therapy-mediated restoration of CD4, which leads to the capacity for a now competent CD8+ T-cell cytolytic response against the reactivation antigen, which may have been impaired prior to ART.

## 4. Discussion

Considering the endemicity of LTBI in India, where the estimates for adult prevalence range from 20 to 60% [3,5] and where TB is the single largest comorbidity associated with HIV infection, it is likely that a large proportion of HIV-associated TB might result from the reactivation of latent infections, although higher baseline CD4 counts (and CD4/CD8 ratios) are also known to reduce the risk of active TB infection [24,25]. However, the incidence of TB soon after HIV acquisition, when CD4+ T-cell depletion is not as severe, together with the demonstration of a lack of LTBI reactivation through CD4 depletion alone in an in vivo model of co-infection, suggests the existence of a complex host TB equilibrium that defines LTBI in the context of HIV [26,27]. In this study, we report on novel and distinct host-associated signatures linked to apparently ameliorated disease progression in PLHIV with latent TB co-infection that in turn suggest the existence of a unique regulatory mechanism that blunts the HIV infection-mediated pathology in the absence of anti-retroviral therapy (ART). The ART-naïve status of our study cohort, together with stratification based on the interferon gamma release assay [IGRA; [28]], the clinical diagnosis of active TB and HIV-1 disease progression markers, permitted a robust and concurrent comparative analysis between co-infect groups for which few data exist [15,25]. At the outset, PLHIV with LTBI (HLTBI+ group) exhibited a disease progression profile comprising the highest CD4+ T-cell counts and CD4/CD8 ratios and viremia similar to the group without co-infection (HLTBI−). On the other hand, PLHIV with active TB co-infection exhibited the highest viral load and lowest CD4/CD8 ratio, confirming the expected advanced HIV disease progression reported earlier [29].

Chronic T-cell activation, a hallmark of both untreated and treated HIV infections [30], including that of CD4+ T cells, which is critical to the control of TB infection [9,31], was evaluated in our study, where a global reduction in activation was uniquely observed in the HLTBI+ group across the CD4+ memory T-cell compartment. Interestingly, an earlier study, restricted to the analysis of CD4+ TB-specific cells [32], had shown a similar and discriminatory signature for HLTBI+ individuals. It is well known that treatment-naïve HIV-1 infection, as well as pulmonary TB infection, is characterized by increased expression of the checkpoint and putative exhaustion marker programmed cell death protein-1 (PD-1) [33]. In our study, we demonstrate a heretofore unreported lowered and unique PD-1 expression profile on memory CD4+ T cells in HLTBI+ individuals that, together with concurrently observed lower T-cell activation, lower intermediate monocyte counts and lower Treg frequencies, demonstrates a relatively (compared to the HTB or HLTBI− groups) less dysregulated immune milieu, which was validated through unsupervised data analysis. 

Additionally, an extensive multi-analyte evaluation of systemic immune mediators, including pro- and anti-inflammatory cytokines, tissue damage and microbial translocation markers, suggested that HLTBI+ individuals, compared to the HTB+ group, exhibited a unique signature that comprised higher circulating IL-10 and IL-17A levels, as well as lower levels of IP-10 and D-dimer. Interestingly, the expression of IL-17 in Th17 cells, essential for tissue repair, regeneration and the maintenance of gut immune homeostasis [34,35,36], is known to be inhibited by PD-1 [37,38], whose expression levels were the lowest on CD4+ T cells, together with those of circulating D-dimer, in the HLTBI+ group. Indeed, an integrative correlational analysis demonstrated a strong negative correlation, uniquely observed in the HLTBI+ group, between the circulating IL-10 and IL-17A levels and PD-1 expression on CD4+ memory T cells. The importance of this relationship in defining the non-reactivated HLTBI+ phenotype was further highlighted through the independent clustering of this group on the basis of the PD-1 expression of the CD4+ CM and TM subsets. An in vivo study by Bucsan et al. [27], exploring mechanisms of LTBI reactivation by SIV co-infection, independently of CD4 depletion, seems to partially corroborate our findings, where reactivated individuals showed reduced IL-17 and PD-1 expression but higher activation in broncho-alveolar lavage T cells.

With respect to increased systemic IL-10 levels in the HLTBI+ group, Gao et al. [39] have reported the functional polarization of macrophages from the M1 to M2 type mediated by DosR Rv1737c (dormancy survival regulator protein of MTB), accompanied by higher (compared to active TB infection) systemic IL-10 expression, in non-HIV-1-affected LTBI+ individuals. We believe that our results demonstrate the preservation of this active immunosuppressive phenotype in HLTBI+ individuals, despite the pathogenic insult associated with HIV-1 acquisition. Indeed, we believe that the preservation of IL-10 and IL-17A levels in HLTBI+ individuals would reduce activation-associated tissue damage and thus exogeneous supplementation in cases where depletion occurs [40,41,42].

While evaluating TB-specific responses to DosR, our observations of apparently unimpaired responses in HIV co-infectioned individuals were contradictory to earlier reports [15,43] where responses in individuals with co-infection were lower than in their respective seronegative control groups. We believe that this may be due to the low number of participants studied in the report by Rakshit et al. [15] and also, in the case of Murray et al. [43], due to the majority (67%) of the HIV+ cohort having a CD4 count lower than 350 cells/µL, which might have significantly impacted the responsiveness. Our results also exhibited responses by a few LTBI− individuals, highlighting the lack of sensitivity of the IGRA assay in the detection of TB-specific responses, especially in the CD4-depleted setting within PLHIV [8,44].

With respect to responses against Rpf, a putative TB reactivation marker [14,45], we report, for the first time in a HIV-TB co-infect cohort, that the highest and most frequent responses (including IFN-γ and TNF-α) observed were in the HTB+ group, suggesting that these individuals may have experienced reactivation as this protein is required for the resuscitation of MTB from dormancy [46]. The opposing TNF-α responses observed in the HLTBI+ and HTB+ groups against DosR and Rpf, respectively, may also reflect the evolution of the disease state from latency to reactivation, which could be validated by future studies in vivo. 

The results of the longitudinal arm of our study highlight the modulation of systemic and TB-specific responses during ART-mediated virological suppression and the accompanying CD4 count rebound, albeit with a persistently higher risk of active TB infection compared to HIV-seronegative settings [47,48]. The restoration of T-cell activation and PD-1 expression, although partial, was observed over six months of ART, together with the lowering of some systemic plasma markers. However, the persistent elevation of memory Tregs, a surrogate of continuing viral reservoir-driven activation, is reminiscent of a phenotype observed by Ganatra et al. [12] in the pulmonary milieu of SIV/LTBI+ rhesus macaques, which exhibited the unhindered development of granulomas, necrotic foci and even bacterial loads after the initiation of ART. An important distinction between our results and this study is the lack of INH prophylaxis therapy (IPT), which was absent in the latter and may have contributed to delaying reactivation in the human setting. Studies with longer (>2 years) follow-up are required to evaluate whether the current regimen of IPT of only six months is sufficient to prevent reactivation over the decades of anticipated ART and unrestored immune dysfunction in PLHIV [30,49,50]. Interestingly, the elevated (compared to HTB+ individuals)—and relatively unperturbed following ART—systemic IL-10 and IL-17A levels observed in our follow-up HLTBI+ cohort may reflect the pulmonary T-cell expression of IL-17A an in vivo protective correlate of LTBI reactivation following SIV co-infection [27]. Thus, preserving these levels in HLTBI+ individuals might present an immunotherapeutic adjunctive strategy for sustained ART [51].

The dynamics of the MTB-specific responses in sensitized (LTBI+) PLHIV following ART initiation in our longitudinal cohort revealed trends of increasing DosR- and Rpf-specific responses by 6 months, in concordance with similar reports and indicative of ART-mediated partial immune reconstitution [52,53,54]. Moreover, one other report from India, focusing on DosR alone, demonstrated higher responses within a year of ART initiation [55]. While our study provides novel data with respect to such responses against a putative reactivation marker, Rpf, we did not notice any major difference in the post-ART responses to DosR or this antigen. A notable exception to the aforementioned increased post-ART MTB-specific responses was the clear and consistently lower production of Mip-1β against both DosR and Rpf observed in our longitudinal cohort. We believe that this may have reflected the IPT- and ART-mediated alteration of the differentiation profiles of MTB-specific T cells in light of the data reported by Saharia et al. in an earlier study [56]. Here, ATT receiving PLHIV with active TB initiating ART showed progressively lower PPD-specific Mip-1β production up to 48 weeks following ART initiation.

The integrative correlation of MTB-specific responses with HIV-1 disease progression markers following ART initiation in our HLTBI+ cohort yielded some interesting and heretofore unreported insights. Prominent among these was the continued strong positive correlation between both DosR- and Rpf-specific TNF-α production by CD8+ T cells and the CD4/CD8 ratio, as well as the strong negative correlation between antigen-specific TNF-α production and either systemic activation or PD-1 expression on T cells. We believe that these were preserved and protective TB-specific responses that may have contributed to the prevention of reactivation prior to and for at least 2 years following ART initiation. Reflective of ongoing immune restoration, we also noted de novo CD8-specific cytolytic responses to both DosR and Rpf that were negatively correlated with systemic PD-1 expression on T cells. It was also noted that some MTB-specific responses, especially cytolytic marker and IFN-γ production by CD4+ T cells, which seemed to be strongly and negatively affected by T-cell PD-1 expression and the viral load prior to ART, were apparently no longer influenced by these parameters within 6 months of therapy initiation. In summation, we have discussed the potential diagnostic relevance of reciprocal TNF-α responses against DosR and Rpf antigens to distinguish between latent and active TB states. These could be developed into prognostic markers for the identification of PLHIV at risk of TB reactivation. The incorporation of such immune monitoring strategies linked to the intensive and targeted administration of anti-TB prophylaxis into clinical disease management would not only aid in early risk stratification but could also inform the development of personalized therapeutic strategies, thereby improving disease management and quality of life for people living with HIV.

Some limitations of our study include the relatively small sample size in the follow-up cohort, as well as the lack of concurrent follow-up (post-ART) of the other HIV-positive groups, which may have provided an interesting comparison of the immune signatures accompanying CD4 rebound with the HLTBI+ group. Moreover, as has been reported before, we also believe that the IGRA positivity may have been influenced by the relatively low CD4 counts, due to which we may have misclassified some LTBI− individuals with latent infections, especially in the case of the HIV-positive groups.

## 5. Conclusions

Our study highlights a unique immunomodulatory environment associated with ameliorated disease progression within LTBI+ individuals with HIV-1 co-infection, both prior to and after the initiation of anti-retroviral therapy. Maintenance of this state probably occurs through the preservation of IL-10 and IL-17A production, lowered PD-1 expression on CD4+ memory T cells and unique MTB-specific signatures, acting as a protective phenotype that mitigates HIV-1-mediated pathogenesis and possibly TB reactivation in these individuals.

## Figures and Tables

**Figure 1 cells-14-01622-f001:**
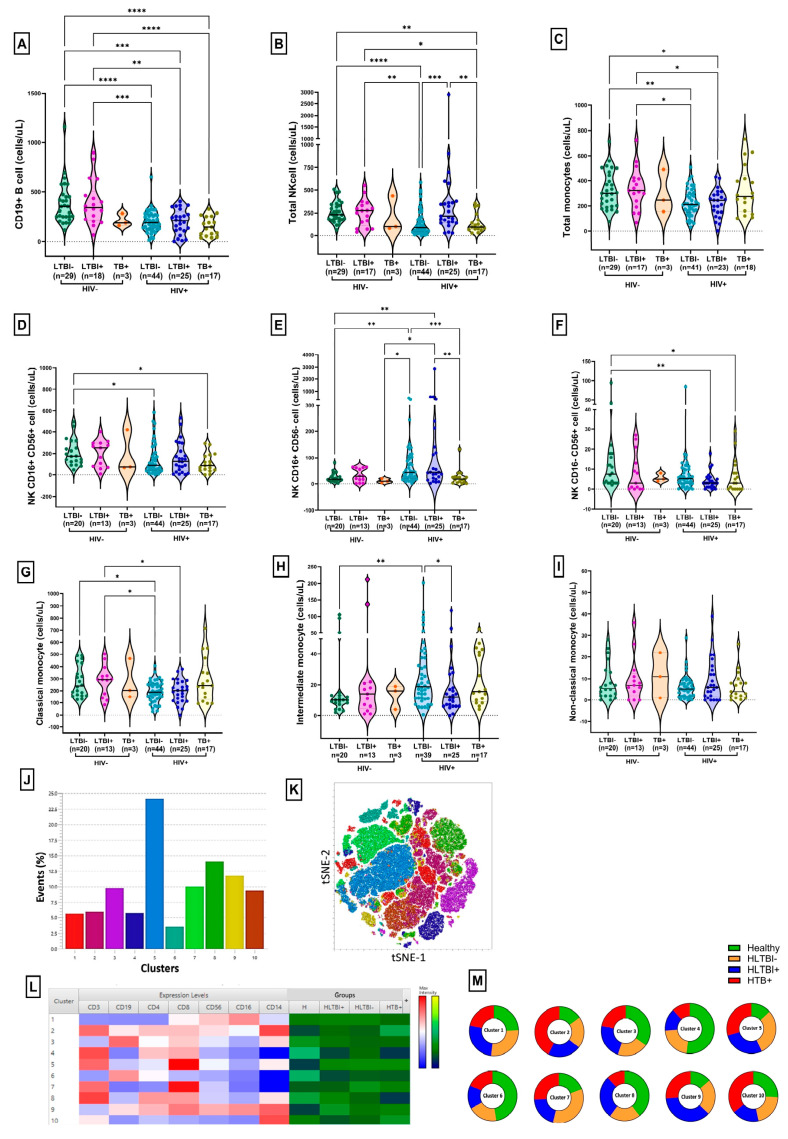
Absolute counts for B, NK cells and monocytes: absolute counts for (**A**) B cells, (**B**) total NK cells, (**C**) total monocytes, (**D**) CD16+ CD56+ NK cells, (**E**) CD16+ CD56- NK cells, (**F**) CD16- CD56+ NK cells, (**G**) classical monocytes (CD14++CD16-), (**H**) intermediate monocytes (CD14++CD16+) and (**I**) non-classical monocytes (CD14+CD16++) across different HIV-seronegative and HIV-1+ groups. Comparisons between groups were performed by Kruskal–Wallis one-way ANOVA non-parametric test (* *p* < 0.05; ** *p* < 0.01; *** *p*< 0.001; **** *p* < 0.0001). Unsupervised dimensionality reduction analysis was performed by t-Stochastic Neighbor Embedding (tSNE) and a graph-based clustering analysis (XShift). (**J**) Bar graphs for frequencies of different clusters, (**K**) tSNE plot for different clusters, (**L**) heatmap presenting differential expression of markers in clusters, with red indicating highest expression and blue indicating lowest expression, with contributions to each cluster for different groups. (**M**) Pie charts representing contributions of each group to different clusters.

**Figure 2 cells-14-01622-f002:**
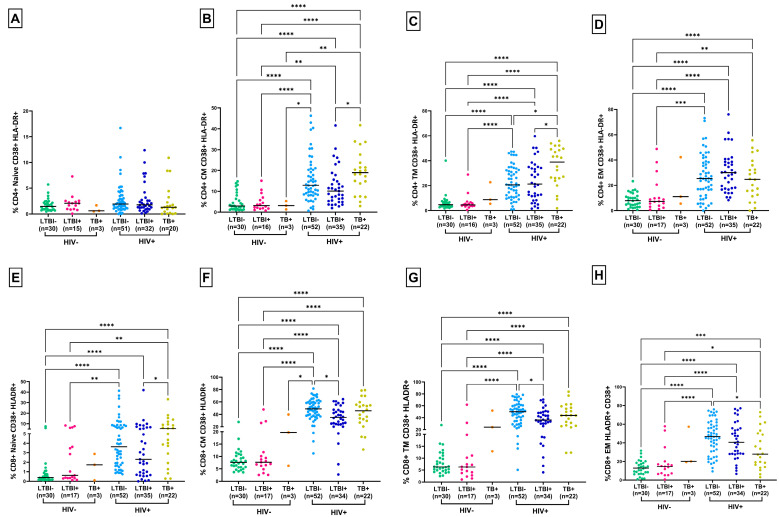
Activation of CD4+ and CD8+ T-cell subsets. Frequency of activated (HLADR+ CD38+) CD4+ T-cell subset—(**A**) naïve, (**B**) central memory, (**C**) transition memory, (**D**) effector memory—and CD8+ T cell subset—(**E**) naïve, (**F**) central memory, (**G**) transition memory, (**H**) effector memory—across different HIV-seronegative and HIV-1-positive groups. Comparisons between groups were performed by Kruskal–Wallis one-way ANOVA non-parametric test (* *p* < 0.05; ** *p* < 0.01; *** *p* < 0.001;**** *p* < 0.0001).

**Figure 3 cells-14-01622-f003:**
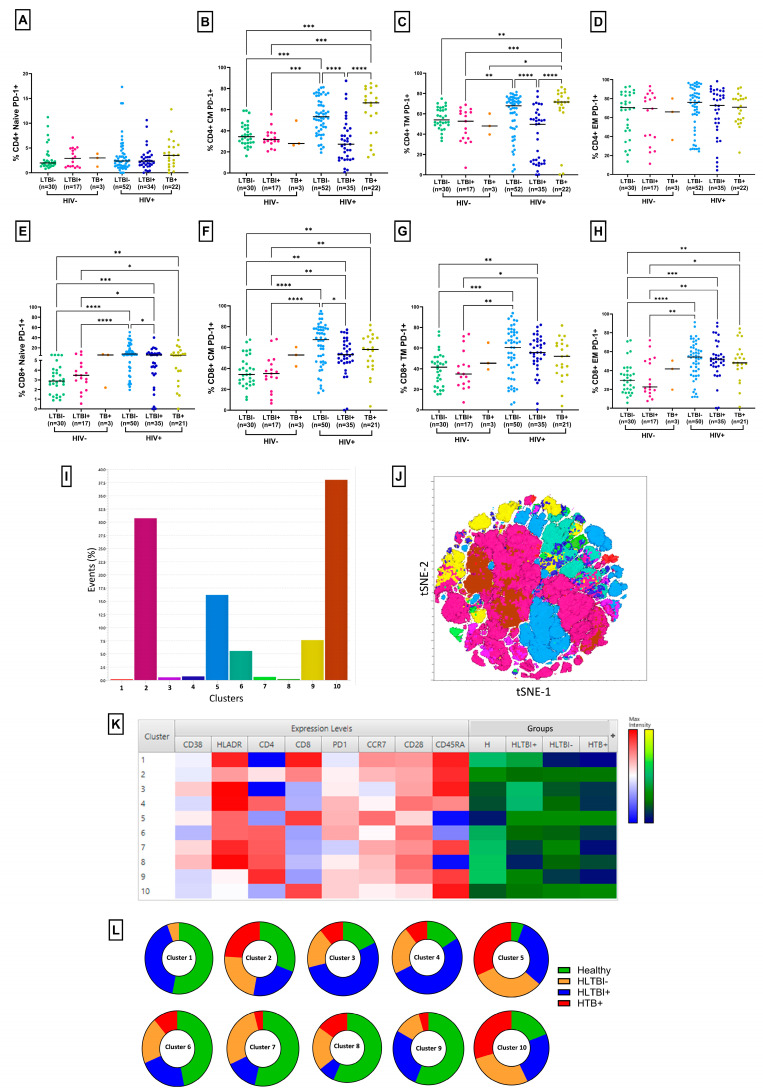
PD-1 expression of CD4+ and CD8+ T-cell subsets: frequency of PD-1+ CD4+ T-cell subset—(**A**) naïve, (**B**) central memory, (**C**) transition memory, (**D**) effector memory—and CD8+ T-cell subset—(**E**) naïve, (**F**) central memory, (**G**) transition memory, (**H**) effector memory—across different HIV-seronegative and HIV-1-positive groups. Comparisons between groups were performed by Kruskal–Wallis one-way ANOVA non-parametric test (* *p* < 0.05; ** *p* < 0.01; *** *p* < 0.001; **** *p* < 0.0001). Unsupervised dimensionality reduction analysis was performed by t-Stochastic Neighbor Embedding (tSNE) and a graph-based clustering analysis (XShift). (**I**) Bar graphs for frequencies of different clusters, (**J**) tSNE plot for different clusters, (**K**) heatmap representing differential expression of markers in clusters, with red indicating highest expression and blue indicating lowest expression, and contributions to each cluster by different groups. (**L**) Pie charts representing contributions of each group to different clusters.

**Figure 4 cells-14-01622-f004:**
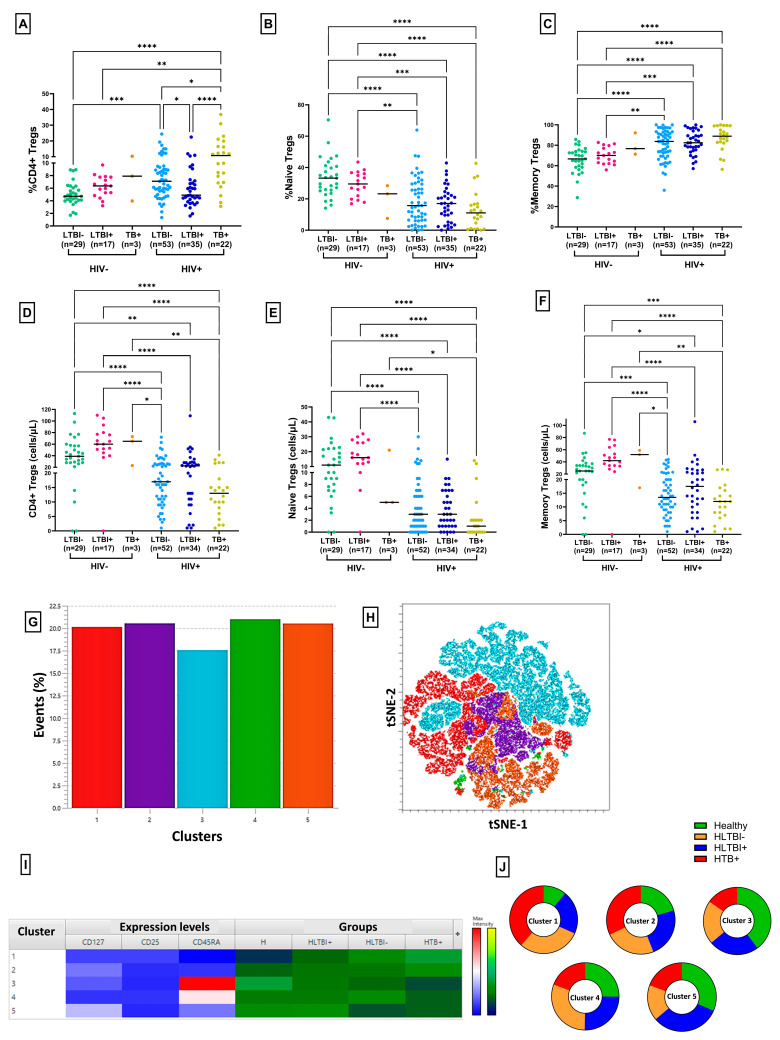
Frequencies and counts of regulatory T cells and subsets: frequencies of CD4+ (**A**) total Tregs, (**B**) naïve Tregs and (**C**) memory Tregs and absolute counts of (**D**) total Tregs, (**E**) naïve Tregs and (**F**) memory Tregs across different HIV-seronegative and HIV-1-positive groups. Comparisons between groups were performed by Kruskal–Wallis one-way ANOVA non-parametric test (* *p* < 0.05; ** *p* < 0.01; *** *p* < 0.001; **** *p* < 0.0001). Unsupervised dimensionality reduction analysis was performed by t-Stochastic Neighbor Embedding (tSNE) and a graph-based clustering analysis (XShift). (**G**) Bar graphs for frequencies of different clusters, (**H**) tSNE plot for different clusters, (**I**) heatmap representing differential expression of markers in clusters, with red indicating highest expression and blue indicating lowest expression, and contributions to each cluster by different groups. (**J**) Pie charts representing contributions of each group to different clusters.

**Figure 5 cells-14-01622-f005:**
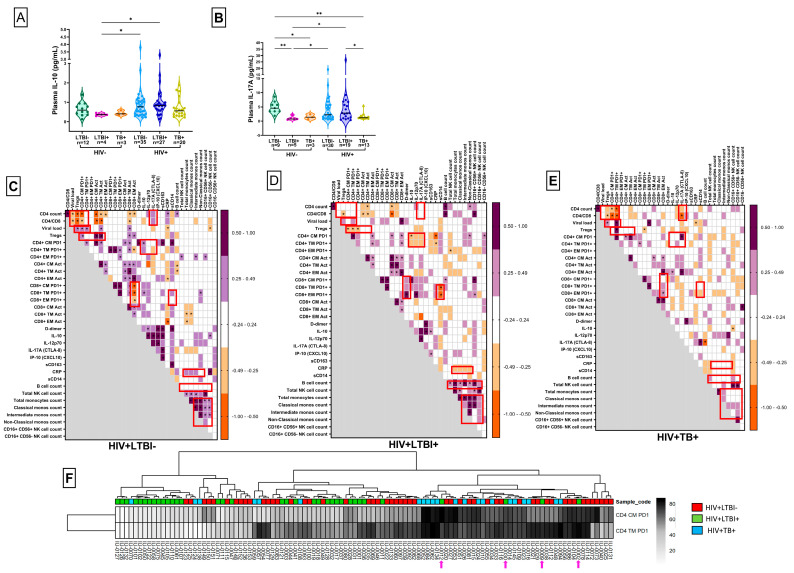
Group-specific pathogenic signatures: levels of plasma (**A**) IL-10 and (**B**) IL-17A in pg/mL. Comparisons between groups were performed by Kruskal–Wallis one-way ANOVA non-parametric test (* *p* < 0.05; ** *p* < 0.01). Heatmap of correlations between systemic immune markers in (**C**) HIV+LTBI− group, (**D**) HIV+LTBI+ group and (**E**) HIV+TB+ group. *p* and *r* values for associations were determined by Spearman’s correlation test. * *p* < 0.05. (**F**) Unsupervised hierarchical clustering analysis for PD-1 expression on CD4+ T cells. The participant IDs marked with pink arrows represent the outliers from HIV+ LTBI+ group.

**Figure 6 cells-14-01622-f006:**
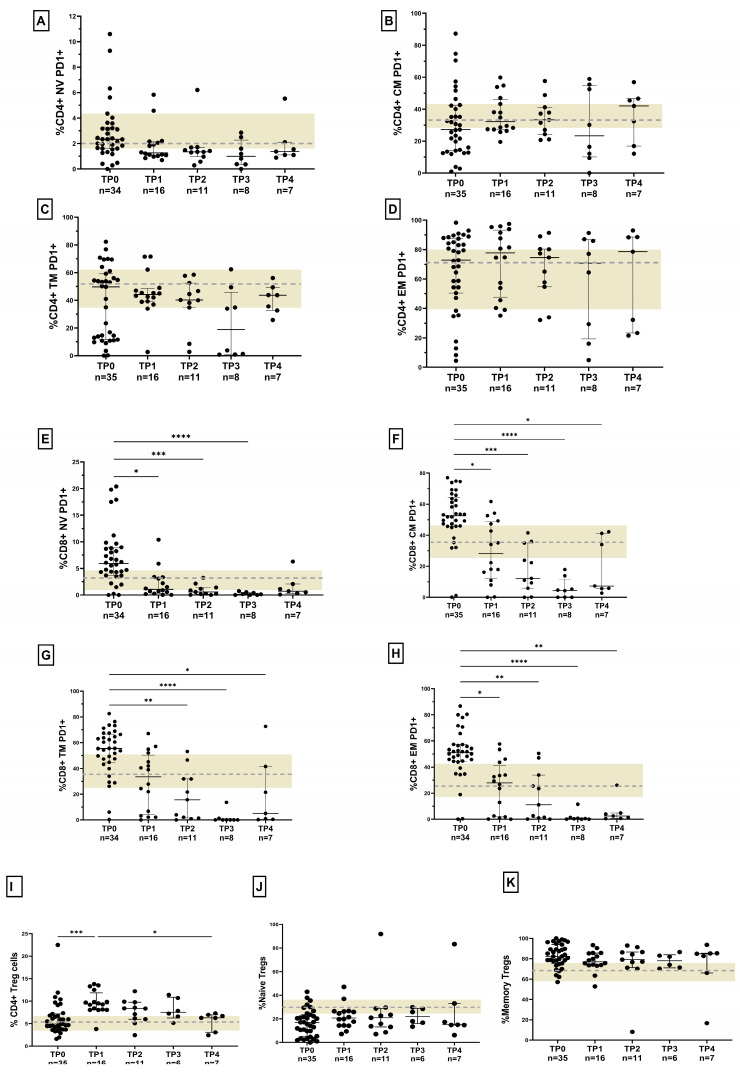
Status of immune restoration following ART initiation in HLTBI+ individuals: unmatched data for frequency of PD-1+ CD4+ T-cell subsets in (**A**) naïve, (**B**) central memory, (**C**) transitional memory and (**D**) effector memory and for CD8+ T-cell subsets in (**E**) naïve, (**F**) central memory, (**G**) transitional memory and (**H**) effector memory. Unmatched data for frequency of (**I**) total Tregs, (**J**) naïve Tregs and (**K**) memory Tregs following initiation of ART at TP1, TP2, TP3 and TP4. The beige-shaded area represents the interquartile range for HIV-seronegative controls, and the grey dotted line represents the median. Comparisons between time points were performed by Kruskal–Wallis one-way ANOVA non-parametric test (* *p* < 0.05; ** *p* < 0.01; *** *p* < 0.001; **** *p* < 0.0001).

**Figure 7 cells-14-01622-f007:**
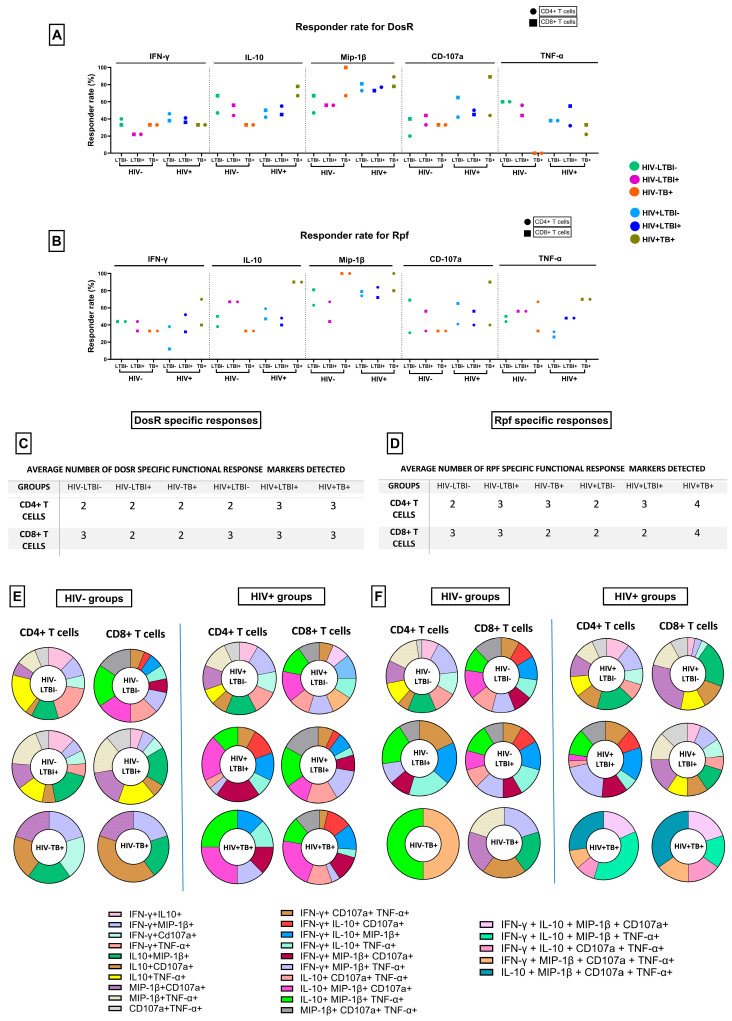
TB-specific intracellular cytokine responses at cross-sectional time points: responder rates for intracellular cytokines IFN-γ, IL-10, Mip-1β, CD107 and TNF-α against. (**A**) DosR antigen, (**B**) Rpf antigen. Table depicting average functional response markers detected for all groups against (**C**) DosR antigen and (**D**) Rpf antigen. Pie charts representing the composition of responses in different combinations of functional responses for all groups against (**E**) DosR antigen and (**F**) Rpf antigen.

**Figure 8 cells-14-01622-f008:**
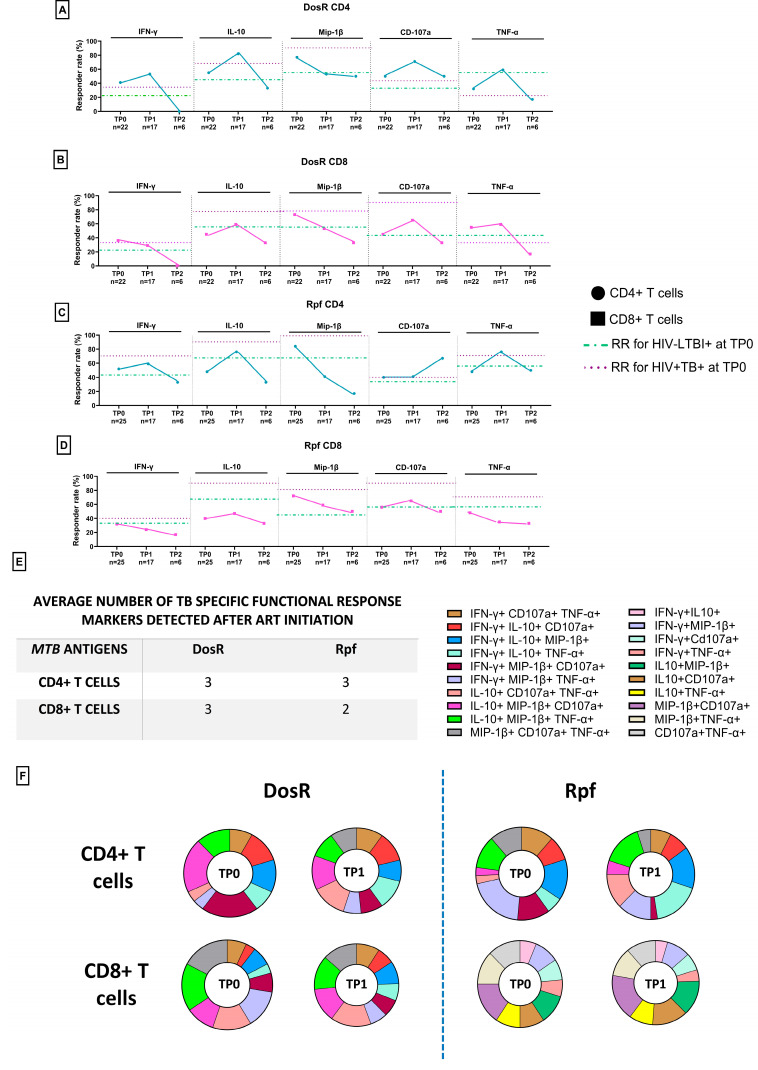
TB-specific intracellular cytokine responses following initiation of ART: responder rates for intracellular cytokines IFN-γ, IL-10, Mip-1β, CD107 and TNF-α at TP0, TP1 and TP2 against (**A**) DosR antigen for CD4+ T cells, (**B**) DosR antigen for CD8+ T cells, (**C**) Rpf antigen for CD4+ T cells and (**D**) Rpf antigen for CD8+ T cells at TP0, TP1 and TP2. (**E**) Table of average TB-specific functional response markers detected for HTBI+ group at TP1. Pie charts representing the composition of responses in different combinations of functional responses for the HLTBI+ group comparing TP0 and TP1 against (**F**) DosR antigen and Rpf antigen.

**Figure 9 cells-14-01622-f009:**
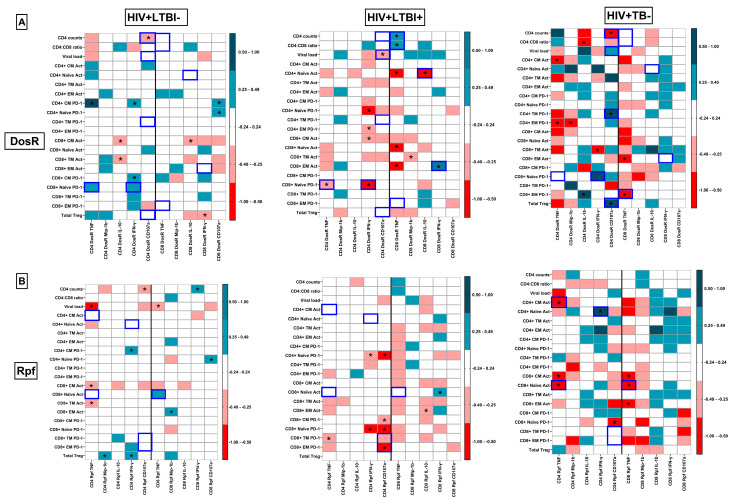
Delineating unique TB-specific pathogenic signatures in HLTBI+ and HTB+ groups: comparison of heatmaps of correlations between systemic immune markers and intracellular functional responses against DosR antigen in (**A**) HIV+LTBI− group, HIV+LTBI+ group and HIV+TB+ group and against Rpf antigens in (**B**) HIV+LTBI− group, HIV+LTBI+ group and HIV+TB+ group. *p* and *r* values for associations were determined by Spearman’s correlation test. * *p* < 0.05.

**Figure 10 cells-14-01622-f010:**
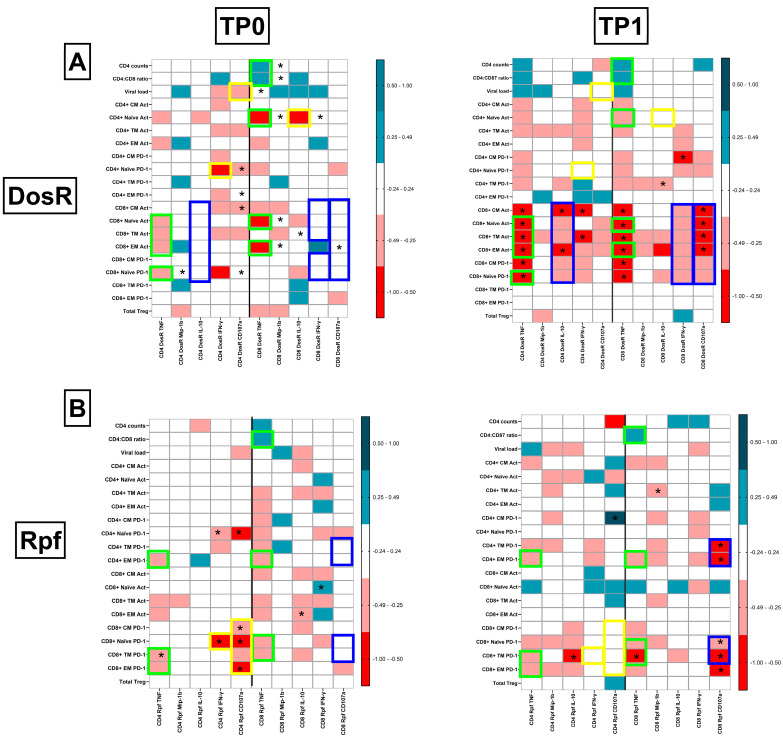
Delineating unique TB-specific pathogenic signatures in HLTBI+ group following initiation of ART at 6 months: comparison of heatmaps of correlations between systemic immune markers and intracellular functional responses in HTLBI+ group at TP0 and TP1 against (**A**) DosR antigen and (**B**) Rpf antigens. *p* and *r* values for associations were determined by Spearman’s correlation test. * *p* < 0.05. Boxes highlighted in green represent signatures that are retained, yellow indicates signatures that are lost, and blue indicates de novo signatures after 6 months of ART.

**Table 1 cells-14-01622-t001:** Abridged table of demographic and disease progression characteristics.

	HIV-1-Positive			HIV-Seronegative		
	HLTBI+	HLTBI−	HTB+	LTBI+	LTBI−	TB+
No. of participants	*n* = 35	*n* = 55	*n* = 22	*n* = 17	*n* = 30	*n* = 3
Gender (F/M)	9/26	20/35	6/16	6/11 *	17/13 *	1/2
Age ^a^	40 (26–59)	37 (18–58)	41 (18–59)	32 (23–48)	29 (22–56)	29 (27–48)
CD4+ T cells (cells/μL) ^a^	362 ^$^ (10–1127)	308 (6–835)	127 (5–455)	969 (552–2478)	837 (468–2019)	707 (286–1623)
CD4/CD8 ^a^	0.39 ^#^ (0.02–1.56)	0.23 (0.01–1.26)	0.12 (0.01–0.5)	1.3 (0.72–2.88)	1.42 (0.7–2.54)	1.26 (0.3–2.45)
HIV-1 viral load (copies/mL) ^a^	63,403 (14–6, 21, 950)	39,905 (22–59, 20, 711)	1, 63, 566 ^@,&^ (67–1, 11, 10, 633)	----	----	----

Note: ^a^ Data represented as median (range). * Age of seronegative controls was significantly lower than that of all infection groups; ^$^ CD4+ T-cell absolute count was higher in HLTBI+ group than HTB+ group, where *p* = 0.0014; ^#^ CD4:CD8 ratio was higher in HLTBI+ group than HTB+ group, where *p* = 0.0030; ^@^ Viral load was highest in HTB+ group than HLTBI− group, where *p* = 0.0273; ^&^ Viral load was highest in HTB+ group than HLTBI+ group, where *p* = 0.0357.

**Table 2 cells-14-01622-t002:** Detailed table of reagent concentrations and volumes for flow cytometry assays.

Assay	Reagent Used and Concentration per Reaction	Reagent Volume and Condition Used
**Absolute cell count (stain/lyse/no wash) [Assay volume 577.5 µL]**		
1. Whole blood	---	50 µL
2. Antibodies (per antibody per reaction)	0.5 µg/mL	Master mix (27.5 µL)
3. FACS lysis solution	1×	450 µL
4. BD liquid counting beads	---	50 µL
**Whole-blood immunophenotyping (stain/lyse/wash) [Assay volume 200 µL]**		
1. Whole blood	---	200 µL
2. Antibodies (per antibody per reaction)	0.5 µg/mL	Master mix for T-cell subset panel (22.5 µL) Master mix for Treg panel (20 µL)
3. RBC lysis, washing with and resuspension in stain buffer	BD FACS lysis buffer (1×) (BD Biosciences, San Jose, CA, USA; catalog no.: 349202) Stain buffer (DPBS + 0.2%FBS)	200 µL
**Intracellular cytokine staining assay [Assay volume 200 µL]**		
PBMCs per reaction	2 × 10^6^ cells/mL	200 µL
2. Stimulation mix (final volume: 10 μL/reaction)		
DosR pool (1 mg/mL)	10 μg/mL	Individual stimulations
b. Rpfpool(1mg/mL)	10 μg/mL	
c. ESAT-6/CFP-10(EC)(250µg/mL)(eachpeptide)	1 μg/mL for each peptide	
d. Phorbol12-myristate13-acetate(PMA)(10mg/mL)	50 ng/mL	Used for positive control
e. Ionomycin(1mg/mL)	1 µg/mL	
f. Brefeldin(Golgiplug)(1000×)	1 μL/mL	Used as protein secretion inhibitor in all reactions
g. Monensin(Golgistop)(1429×)	0.7 μL/mL	
**3. Staining reaction: 200 μL**		
LIVE/DEAD^TM^ Fixable Violet Dead Cell Stain (200×)	1× per reaction	Used in all reactions
b. Surfacestaining(perantibodyperreaction)instainbuffer	0.5 µg/mL	Master mix (17.5 μL)
c. FixationandpermeabilizationwithBDCytofix/Cytoperm	200 μL/reaction	Used in all reactions
d. Intracellularstaining(perantibodyperreaction)in1×BDpermwashfollowedbywashingandresuspensioninstainbuffer	0.5 µg/mL	Master mix (20 μL)

## Data Availability

All data are available in the manuscript and the Appendix A; any additional data required can be provided upon request to the corresponding author.

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
