# Peer review of "Protective Immune Signatures Associated with Latent TB Infection in PLHIV—Insights from an Integrative Prospective Immune Monitoring Study"

_cells, 2025, doi:10.3390/cells14201622_

Round 1
Reviewer 1 Report
Comments and Suggestions for Authors Bhowmick etal presents a study examining immunological signatures in ART-naïve people living with HIV (PLHIV) with latent TB (LTBI+), active TB (ATB+), or no TB co-infection. The authors provide insights into how a distinct immunomodulatory profile in the LTBI+ group—characterized by lower T cell activation, reduced PD-1 expression, and preserved TB-specific TNF-α responses—may play a protective role against both HIV pathogenesis and TB reactivation. The integration of systemic and antigen-specific (DosR/Rpf) immune profiling with longitudinal ART-mediated immune reconstitution data is commendable.I have several comments:
Major comments:
-
The mauscript should provide a more detailed explanation of the rationale behind the sample size selection for each subgroup and make sure that the sample sizes were sufficient for support the conclusion.
-
The number of participants retained at each time point (TP1 to TP4) needs to be clearly specified.
-
The methods should be provided a lot numbers for key reagents used in ICCS assays, particularly the DosR and Rpf antigens, as well as antibodies.
-
Given that the DosR regulon includes around 48 genes, it would strengthen the manuscript to provide a clear justification for the selection of the four proteins studied
-
Regarding the Methods section, the lot numbers of the reagents should be provided, particularly for the DosR and Rpf antigens, as well as antibodies.
-
The statistical analysis requires improvement. For example, Table 1 should include p-values to supprot comparisions across groups.
-
The figure intepretation should be improved.
-
The IL-10 and IL-17A plasma data in Supplementary Figure S6 should be moved into the main results section 3.4.
minor comments:
-
Clarify if PD-1 expression in CD8+ T cells varied across different subsets, or if it followed a consistent pattern like what was seen in CD4+ T cells.
-
The english should be improved for clearer interpretation of the results.
Reviewer 2 Report
Comments and Suggestions for Authors
Review Comments
This manuscript highlights an immunomodulatory phenotype conferred by latent TB infection in PLHIV, whose preservation may provide strategies to mitigate TB reactivation. Overall, this manuscript is beautifully written, and there are only some problems needed to be addressed before being published in Cells.
- The hyphen between “Table” and “1” in Line 13 should be deleted.
- The excess space between the numerical figure number and the alphabetic figure number should be deleted (e.g., the space between “Figure 2” and “B” in Line 36).
- Some words in the figures are too small to be clearly read (e.g., Fig. 3I, 4G, 5D, etc.).
- The colon in the heading of Section 3.4 should be removed.
- The first letter of the phrase “figure 9” in Section 3.10 should be capitalized.
- The abbreviation list in Page 24 is not necessary in this manuscript for there are not many abbreviations in this manuscript.
- The format of the references should be further improved.

Reviewer 3 Report
Comments and Suggestions for Authors
This manuscript presents an original and comprehensive investigation of the immune phenotype and function in people living with HIV (PLHIV) co-infected with latent tuberculosis infection (LTBI) and active TB, both before and after antiretroviral therapy (ART) initiation. The article is well-written and presents original and valuable findings; however, it requires certain revisions and additions before it can be considered suitable for publication.
Major comments
- In the Introduction, please clarify the sentence (Lines 53-54)“… a population that accounts for about 26% of the global deaths caused among people with and without HIV” to specify whether this relates specifically to TB deaths or HIV/TB co-infection mortality concentrated in India.
- There is some discrepancy in the group size. Section 2.2 states that 82 HIV-1-infected ART-naïve and 50 HIV sero-negative individuals were recruited, and that the HIV+ group was further divided into LTBI+, Active TB+, and non-co-infected. However, in other parts of the paper (e.g., in the Results section 3.1 and the Abstract), the number of HIV+ individuals is reported as 112, not 82. There is thus a difference of 30 participants that needs clarification.
- The manuscript does not specify if the key immunological tests, such as flow cytometry, ELISA, and IGRA, were performed with technical replicates. Details on quality control measures, including the number of replicates or other strategies ensuring assay reliability, are not clearly described in the Methods section.
- No clear indication of statistical significance in the text description. Although in many places the authors refer to statistics, they rarely provide specific p-values directly in the text.
- The discussion could be improved by detailing how the obtained results could be applied to clinical practice, such as diagnostic biomarkers, risk stratifiers, or therapeutic targets to avoid TB reactivation.
- The authors could provide more information on how their immunological findings relate to current HIV treatment strategies and potential integration into patient management.
Minor comments
- Please ensure consistent notation of units throughout the manuscript (e.g., μL instead of uL).
- Please correct temperature formatting to use “37°C” rather than “370C” (Line 40).
- Some of the axes labels on the figures (e.g., Figure 1, 7A, 7B, especially within individual panels) are unclear and require revision to ensure that all axes are fully legible and easily interpretable.
Reviewer 4 Report
Comments and Suggestions for Authors
The authors examined a cohort of HIV-infected patients for immune signatures associated with latent tuberculosis infection. Fifty HIV-seronegative and 112 HIV-infected patients undergoing antiretroviral therapy were examined for latent or active tuberculosis infection and followed up for two years. Using extensive multiparametric flow cytometry, changes in various leukocyte populations were detected.
The materials and methods are described in detail and are easy to follow. The separation of cytometry into absolute cell counting, differentiation, and intracellular measurement is unusual. Unfortunately, most of the concentration and volume data are missing from the protocols.
The recruited patients are presented in detail, followed by the results for the individual cell populations.
Conclusions and treatment recommendations are derived from the results and discussed later.
The references are correct and the supplementary material is helpful.
Round 2
Reviewer 1 Report
Comments and Suggestions for Authors
I have carefully reviewed the revised version of your manuscript. I am pleased to confirm that all the comments have been fully and appropriately addressed in this new version. I think the revised manuscript is ready for acceptance.